# Bacterial Nanocellulose Fortified with Antimicrobial and Anti-Inflammatory Natural Products from *Chelidonium majus* Plant Cell Cultures

**DOI:** 10.3390/ma15010016

**Published:** 2021-12-21

**Authors:** Sylwia Zielińska, Adam Matkowski, Karolina Dydak, Monika Ewa Czerwińska, Magdalena Dziągwa-Becker, Mariusz Kucharski, Magdalena Wójciak, Ireneusz Sowa, Stanisława Plińska, Karol Fijałkowski, Daria Ciecholewska-Juśko, Michał Broda, Damian Gorczyca, Adam Junka

**Affiliations:** 1Department of Pharmaceutical Biology and Biotechnology, Division of Pharmaceutical Biotechnology, Wroclaw Medical University, 50-556 Wroclaw, Poland; sylwia.zielinska@umw.edu.pl; 2Department of Pharmaceutical Biology and Biotechnology, Division of Pharmaceutical Biology and Botany, Wroclaw Medical University, 50-556 Wroclaw, Poland; 3Pharmaceutical Microbiology and Parasitology, Wroclaw Medical University, 50-556 Wroclaw, Poland; karolina.dydak@umed.wroc.pl (K.D.); adam.junka@umw.edu.pl (A.J.); 4Department of Biochemistry and Pharmacogenomics, Faculty of Pharmacy, Medical University of Warsaw, 02-097 Warszawa, Poland; monika.czerwinska@wum.edu.pl; 5Centre for Preclinical Research, Medical University of Warsaw, 1B Banacha Street, 02-097 Warsaw, Poland; 6Department of Weed Science and Tillage Systems, Institute of Soil Science and Plant Cultivation State Research Institute, 50-540 Wrocław, Poland; m.dziagwa@iung.wroclaw.pl (M.D.-B.); m.kucharski@iung.wroclaw.pl (M.K.); 7Department of Analytical Chemistry, Medical University of Lublin, 20-093 Lublin, Poland; magdalena.wojciak@umlub.pl (M.W.); ireneusz.sowa@umlub.pl (I.S.); 8Department of Inorganic Chemistry, Wroclaw Medical University, 50-556 Wrocław, Poland; stanislawa.plinska@umed.wroc.pl; 9Department of Microbiology and Biotechnology, Faculty of Biotechnology and Animal Husbandry, West Pomeranian University of Technology, 70-311 Szczecin, Poland; kfijalkowski@zut.edu.pl (K.F.); daria.ciecholewska@zut.edu.pl (D.C.-J.); michal.broda@zut.edu.pl (M.B.); 10Pomeranian-Masurian Potato Breeding Company, 76-024 Strzekęcino, Poland; 11Faculty of Medicine, Lazarski University, 02-662 Warszawa, Poland; damian.gorczyca@lazarski.pl

**Keywords:** bacterial nanocellulose, *Chelidonium majus*, cell culture carrier, phenolic compounds, isoquinoline alkaloids, microbial pathogens

## Abstract

In this work we developed a bi-functional Bacterial-Nano-Cellulose (BNC) carrier system for cell cultures of *Chelidonium majus*—a medicinal plant producing antimicrobial compounds. The porous BNC was biosynthesized for 3, 5 or 7 days by the non-pathogenic *Komagataeibacter xylinus* bacteria and used in three forms: (1) Without removal of *K. xylinus* cells, (2) partially cleaned up from the remaining *K. xylinus* cells using water washing and (3) fully purified with NaOH leaving no bacterial cells remains. The suspended *C. majus* cells were inoculated on the BNC pieces in liquid medium and the functionalized BNC was harvested and subjected to scanning electron microscopy observation and analyzed for the content of *C. majus* metabolites as well as to antimicrobial assays and tested for potential proinflammatory irritating activity in human neutrophils. The highest content and the most complex composition of pharmacologically active substances was found in 3-day-old, unpurified BNC, which was tested for its bioactivity. The assays based on the IL-1β, IL-8 and TNF-α secretion in an in vitro model showed an anti-inflammatory effect of this particular biomatrix. Moreover, 3-day-old-BNC displayed antimicrobial and antibiofilm activity against *Staphylococcus aureus*, *Pseudomonas aeruginosa* and *Candida albicans*. The results of the research indicated a possible application of such modified composites, against microbial pathogens, especially in local surface infections, where plant metabolite-enriched BNC may be used as the occlusive dressing.

## 1. Introduction

The bacterial nanocellulose (BNC) properties have attracted world-wide attention from the biomedical and biotechnological industries. Thanks to its structure, BNC was used as a cell carrier for various technological processes. The cellulose is considered one of the most ubiquitous compounds on Earth [1]. In comparison to plant cellulose, BNC displays higher purity, degree of polymerization, water-related (swelling and release) properties and crystallinity index. So far, it is applied as a component of wound dressing, food additive, drug carrier, electro-wires, environment-friendly garments (jackets, purses, shoes) and others [2,3,4,5]. BNC is produced by a range of microorganisms, of which *Komagataeibacter xylinus* is considered the most efficient [6]. The removal of *K. xylinus* from BNC includes subsequent stages of alkaline lysis and multiple rinsing. After cleansing, the cell-free BNC displays the high biocompatibility and lack of cytotoxic/allergic effect in contact with live tissues [7].

To our best knowledge, BNC has not yet been applied, even in a proof-of-concept study, as a carrier for plant cells producing pharmacologically active metabolites.

Therefore, our challenge was to introduce alkaloid and polyphenolic-secreting plant cells into the biocellulose carrier, in order to obtain a ready-to-use matrix containing metabolites of medicinal properties. The research hypothesis was based on three assumptions: (i) Plant cells may adhere to the NBC; (ii) such attached plant cells retain ability to produce pharmacologically active metabolites; and (iii) various types of NBC cleansing correlate with various levels of metabolites production.

For the verification of our hypothesis, we chose undifferentiated cells of *Chelidonium majus* L. (Greater celandine) as an experimental model for the plant BNC composite system enriched in the bioactive metabolites. Our previous results shown that the *C. majus* callus cells were able to produce several classes of isoquinoline alkaloids [8]. Among isoquinoline alkaloids, benzophenanthridine derivatives such as chelidonine, sanguinarine and cheletythrine, and also protopine and protoberberine derivatives were found to present strong biological activities of wide spectra, which have been extensively presented in our recently published review article [9] as well as in two experimental studies [10,11].

Chelidonine exhibits effects on the central nervous system similar to those of morphine but weaker, and spasmolytic effects on smooth muscles similar to those of papaverine but also weaker. It is also a spindle poison [12,13,14,15]. Chelerythrine strongly irritates the skin and mucous membranes, benumbs the central nervous system and acts as a local anesthetic [16,17]. Sanguinarine exhibits a vasorelaxant effect and inhibitory effect on smooth muscle contractions. It is also an inhibitor of acetylcholiesterase and 5-lipoxygenase and it presents antimicrobial activity [17,18,19]. Protopine can act as an analgesic and inhibits histamine H1 receptors and thrombocytes aggregation [20,21]. Coptisine was found to present neuroprotective, cytotoxic and inhibitory effects on monoamine oxidase, and cardioprotective bioactivities. Its derivatives were tested as a potential anti-ulcerative colitis agents. According to the latest data, coptisine exhibits much higher cytotoxic effect compared to other alkaloids found in *C. majus* [11,22,23,24,25]. All of them have been extensively studied; however, new data are appearing on their mechanisms of action. The second large group of specialized metabolites in *C. majus* cells are such polyphenolic compounds as chlorogenic acid, trans-caffeic acid and quercetin. The first two are classified as hydroxycinnamic acids, and the third one is a flavonoid.

Generally, these substances can be highly active towards oxidation. Chlorogenic acid may also reduce blood pressure and has antibacterial activity [26,27,28].

Being aware of increasing antimicrobial resistance towards antibiotics and antiseptics [29], resulting in a high clinical demand of new antimicrobials and the way of its administration, in a present proof-of-concept experiment, we investigated the possibility of culturing *C. majus* cells on a BNC carrier. We tested three types of BNC: unpurified polymer containing live *K. xylinus* cells (NC), BC with killed (by standard autoclaving) but not removed *K. xylinus* cells (H_2_O) and BC with killed and removed *K. xylinus* cells (NaOH). The chemical characteristics of the obtained complex matrices and their components in terms of plant specialized metabolites presence were investigated using liquid chromatography–mass spectrometry (LC-MS). The effects of selected BNC composites on the immune system cells and microbial cells in vitro were tested as well.

## 2. Materials and Methods

*C. majus* cells were cultured in vitro on BNC matrices submerged in liquid media. After the culture period, cellulose was enzymatically digested and purified. The phytochemical profile was determined using HPLC. Cellulose matrices together with intertwined plant compounds were also examined using scanning electron microscopy (SEM) and the antimicrobial potential of the BNC was evaluated against a set of planktonic and biofilm-forming pathogens, *Staphylococcus aureus* ATCC 6538, *Pseudomonas aeruginosa* ATCC 15,442 and *Candida albicans* ATCC 10231. Additionally, the effect of plant metabolite-rich cellulose on neutrophil cytokines released by LPS-stimulated cells was performed, and its cytotoxicity was evaluated.

### 2.1. Plant Cell Culture General Procedures

*C. majus* cell cultures were obtained from root explants according to the modified procedure previously described by Zielińska et al. (2018) [8]. After 12 months of culture on solid MH3 media supplemented with 2,4-Dichlorophenoxyacetic acid [2.5 µM] 1-Naphthaleneacetic acid [0.5 µM] Kinetin [1.0 µM] and callus fragments (0.5 g) were dispersed by vortexing to contain mostly individual cells and suspended in 100 mL of liquid MH3 phytohormone-free medium. MH3 medium was prepared according to Morard et al. 1999 [30]. 

### 2.2. Cell Viability Assay

Plant cells viability was estimated after two and four weeks of culture in MH3 liquid medium. The quantification of live plant cells was performed using Plant Cell Viability Assay Kit (Sigma, Roedermark, Germany). The stains were diluted 100× in water to obtain 10× stain solution. The 500 µL of cell suspension was diluted 2× in water. Ten microliters of diluted stains were next introduced to 90 µL of cell suspension. Such samples were incubated at room temperature for 2 min. After incubation, the 20 µL of dyed cell solution was placed onto microscope slide and secured with square-shape coverslip. Subsequently, the pictures of dyed cells were captured using Lumascope 620 (Etaluma, Carlsbad, CA, USA) and magnification × 20 with Exmax: 494 nm; Emmax: 518 nm for fluorescein acetate and Exmax: 536 nm; Emmax: 617 nm for Propidium iodide. The total 24 fields of vision were recorded. To count the cells, the pictures were processed using ImageJ (National Institutes of Health, Bethesda, MD, USA) software. Type of picture was changed into 32-bit black and white, then threshold was established to differentiate objects from noise; aggregates were divided using “watershed” option, and number of object representing plant cells was counted using “analyze particles” option. The surface of field of vision was 0.2 mm^2^; 20 µL of suspension was distributed under the surface of 81 mm^2^; the total number of cells/1 mL was counted using formula: average number of cells counted from 24 fields of vision × 81/0.2 × 50.

### 2.3. The BNC Biosynthesis and Carrier’s Characteristics

#### 2.3.1. Biosynthesis

The BNC was biosynthesized in a sterile glass jar of 50 mm diameter by reference strain of *K. xylinus* (Deutsche Sammlung von Mikroorganismen und Zellkulturen- German Collection of Microorganisms and Cell Cultures DSM 46604). The *K. xylinus* suspension (2 × 10^5^ CFU/mL) obtained from a 7-day-long culture was used to inoculate the Herstin–Schramm medium (glucose (2 *w*/*v*%; POCH, Gliwice, Poland), yeast extract (0.5 *w*/*v*%; Graso, Starogard, Poland), bacto-pepton (0.5 *w*/*v*%; Graso), citric acid (0.115 *w*/*v*%; POCH), Na_2_HPO_4_ (0.27 *w*/*v*%; POCH), MgSO4·7H_2_O (0.05 *w*/*v*%; POCH), bacteriological agar (2% *w*/*v*; Graso) and ethanol (1 *v/v*%; POCH). The BNC biosynthesis was carried out under a stationary condition at 28 °C for 3, 5 or 7 days. The obtained BNC carriers in a form of flat cylinders (Figure 1) were harvested from the medium after cultivation and:(a)Applied in the intact form (containing *K. xylinus* cells), such carriers were later referred to as the “NC”;(b)Introduced to 25mL of sterile water and subjected to standard process of autoclaving (Vapour Line 135 M device, VWR, Radnor, PA, USA) at 134 °C; such carriers were later referred to as the “H_2_O”;(c)Purified by treatment with 0.1 M NaOH (POCH, Poland) at 80 °C for 90 min to remove the bacterial cells and medium components. The purified BC samples were washed in sterile water until neutral pH value of washing water was reached. Such carriers were later referred to as the “NaOH”.

The weight of wet BC carriers was measured using laboratory balance (Pioneer Model PA 114CM/1, Ohaus, Parsippany, NJ, USA).

**Figure 1 materials-15-00016-f001:**
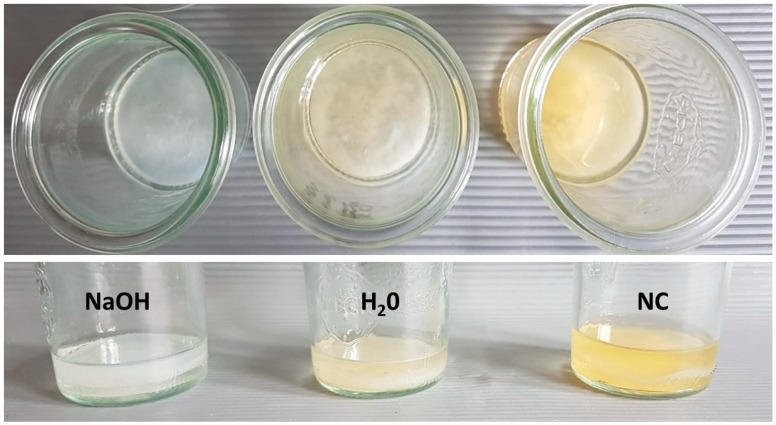
NBC cultured in glass vessel took form of cylinder of 50 mm diameter. The upper panel shows the top, while the lower panel the side plane of 3 days old NBC carriers subjected to *K. xylinus* cells full removal (NaOH, alkaline lysis) or to process of auto-claving in water (H_2_O), or NBC which was not subjected to any cleansing process (not-treated, NC).

#### 2.3.2. The BC Carrier Characteristics

To measure the swelling properties, the dried BC samples were immersed in distilled water and weighed every 1 min until a constant weight of the wet sample was achieved. The porosity-related parameters of BC membranes were evaluated by N_2_ adsorption/desorption measurements. The N_2_ adsorption/desorption isotherms at 77 K were measured using a Micromeritics ASAP 2010 M instrument (Micromeritics, Norcross, GA, USA) and the specific surface area was calculated by the BET (Brunauer–Emmett–Teller) method. The pore volume and pore diameter were calculated by the BJH (Barrett–Joyner–Halenda) method. The mechanical strength was determined using the MTS Synergie 100^®^ machine (MTS Systems Corp, Eden Prairie, MN, USA). The analyses were carried out at a speed of 1 mm/min at room temperature in six repeats. The viability of BC-producing bacterial cells was determined after digestion of the cellulose pellicles with cellulase, using AlamarBlue^®^ assays (ThermoFisher, Waltham, MA, USA). For the digestion, the cellulose pellicles were washed in distilled water, transferred to 5 mL of the cellulase solution in citrate buffer (0.05 mol/L, pH 4.8), and incubated with shaking for 24 h at 30 °C. Next, the samples consisting of cell suspensions from pellicles enzymatic hydrolysis were centrifuged for 20 min at 3300× *g*. The obtained pellets were washed in phosphate buffered saline (PBS, Sigma-Aldrich, St. Louis, MO, USA) and again centrifuged at the same conditions and finally restored to the original volume with PBS. The bacterial suspensions (200 μL) were then transferred into wells of 96-well fluorescence microtiter plates (Becton Dickinson, Franklin Lakes, NJ, USA) and 20 μL of AlamarBlue^®^ was added, followed by 1 h incubation for AlamarBlue at 30 °C in the dark. The fluorescence was measured using microplate fluorescence reader (Synergy HTX, Biotek, Winooski, VT, USA) at wavelengths 540 nm excitation and 590 nm emission. As a blank, sterile PBS was used. The correlation of viability and cell number was performed with use of previously prepared calibration curve.

### 2.4. Plant Cell Cultures on BNC Matrices

The BNC carriers of NC, H_2_O and NaOH type obtained after 3, 5 and 7 days of bacterial culture (described in 4.3) were used in the experiments. Each of the cellulose carrier types were used in six repetitions. Pieces of BNC in a form of flat cylinders were submerged in MH3 liquid medium (10 mL) and autoclaved (121℃, 20 min; MLS-3781L, Panasonic, Etten-Leur, the Netherlands). Four-week old *C. majus* suspension cell cultures were transferred (5 mL) into glass jars (140 mL of volume) containing one of the three types of flat cellulose cylinders submerged with 10 mL of fresh liquid MH3 culture medium (see the Appendix A—Vertical cross-section BNC containing *C. majus* cells. The blue areas show regions containing cells—mainly on the top of BNC carrier and on the side parts of BNC carrier. The middle and bottom parts are dye-free, suggesting lack of live *C. majus* cells in these areas. (Appendix A—The processes schematic of the *C. majus* cells cultured on Bacterial Nano-Cellulose (BNC) matrix). For the results of the total number of cells/1 mL cultured for two and four weeks in Erlenmeyer flasks, (see sub-Section 2.2. “Cell viability assay” and Appendix A, as well as the Results section). *C. majus* cells were cultured on BNC matrices in glass jars on rotary shakers placed in growth chamber at 25 ± 2 °C in dark for 14, 21 and 28 days. The shakers’ (SKC 6200, Lab Companion, Daejeon, Korea) speed was set at 85 rpm, and the amplitude was 30 mm. After the incubation, the plant cells-enriched-BNC was examined under scanning electron microscope (SEM) or digested using cellulase (described in 2.3.3.). The digested samples were then used for the phytochemical analysis and samples from 3-days BNC for bioactivity assays.

### 2.5. Visualization of C. majus Cells on BNC Carriers

The samples (BNC carriers with *C. majus* cells adhered to it) were gently cleansed in PBS (Sigma-Aldrich, Darmstadt, Germany) buffer as it was described in; fixed in glutaraldehyde (POCH, Wroclaw, Poland) and dried in a critical point dryer EM CPD300 (Leica Microsystems, Wetzlar, Germany). Subsequently, the samples were subjected to sputtering with Au/Pd (60:40) using EM ACE600, Leica sputter (Leica Microsystems, Wetzlar, Germany). The sputtered samples were examined using a scanning electron microscope (SEM, Auriga 60, Zeiss, Jena, Germany). Additionally, the porosity-related parameters (pore diameter and pore volume) were determined by N_2_, adsorption/desorption method (ASAP 2010 M, Micromeritics, Norcross, GA, USA).

### 2.6. Phytochemical Analyses

The composition of polyphenolic compounds and isoquinoline alkaloids contained in all types of BNC carries were analyzed using LC–ESI–MS/MS (Shimadzu, Kyoto, Japan). The qualitative and quantitative analysis as well as the source of the reference substances are described in our previously presented method by Zielińska et al. 2020a [11].

### 2.7. Bioactivity Assays

#### 2.7.1. TNF-α, IL-8 and IL-1β Production by PMNs

The effect of 3-day-old BNC cellulose containing non-removed *K. xylinus* bacterial cells and *C. majus* cells on cytokines (TNF-α, IL-8 and IL-1β) secretion by human neutrophils (PMNs) as well as its influence on PMNs viability and cytotoxicity were performed according to the previously reported methodology [11]. The supernatant obtained by cellulase lysis (described in 2.3.3) was treated as 100% 3-day-old plant cell enriched BNC and the dilutions were made with RPMI 1640 (50 and 25%).

#### 2.7.2. Anti-Microbial Tests

The BNC carriers after cultivation of *C. majus* cells on them, were subjected to enzymatic lysis with use of cellulase solution in citrate buffer and, after BNC dissolution, to centrifugation (3300× *g*). The obtained supernatant was applied for antimicrobial tests. The following strains from ATCC (Manassas, VA, USA) were used: *S. aureus* 6538, *P. aeruginosa* 15442; *C. albicans* 10231. For assessment of Minimal Microbiocidal Concentration, the microbial suspensions at a density of 0.5 McFarland (MF) (densitometer Densitomat II, BioMerieux, Warszawa, Poland) were diluted to the 10^5^ cfu/mL. The density of the cellular suspension was confirmed by the quantitative culturing on appropriate agar plates (Columbia Agar for *S. aureus*; McConkey Agar for *P. aeruginosa*; Sabouraud Agar for *C. albicans*). All agar plates were purchased from Biomaxima (Lublin, Poland). The microbial suspensions (100 µL) was introduced to the well of 96-well plate (VWR, Radnor, PA, USA). Next, the metabolite-containing supernatant was introduced to the wells of plate in such a manner that the highest concentration of supernatant was 50% (*v/v*) and the lowest was 1% (*v/v*) in the applied experimental setting. Next, the plates were subjected to continuous mechanical shaking at 350 rpm (Mini-shaker PSU-2T, Biosan SIA, Riga, Latvia) to prevent from settling the cells down for 24 h/37 °C. All experiments were performed in six replicates. In case three of them, 20µL of 1% (*w*/*v*) TTC (2,3,5-triphenyl-tetrazolium chloride, AppliChem GmbH, Damstadt, Germany) solution in TSB was added, incubation has proceeded for 2 h. The aim was to observe change of colorless TTC to red formazan, what confirms presence of live, metabolically active cells in experimental setting. In case of three of replicates, the well contents were transferred to the 10 mL of sterile TSB medium (Biomaxima, Lublin, Poland) and incubated for 24 h/37 °C. Then, the 1 mL of TSB medium was introduced to the three appropriate agar plates (Columbia Agar for *S. aureus* or McConkey Agar for *P. aeruginosa* or Sabouraud Agar for *C. albicans*) and subjected to another 24 h/37 °C incubation. The growth of microbial cells was assessed visually. The accordance between colorimetric and quantitative culturing was performed. Because of lack of growth of microorganisms in specific concentrations of supernatant after introduction to sterile TSB medium, the effect was described as of “microbiocidal” (the presence of *C. albicans* among tested species excluded use of word “bactericidal”). In case of Minimal Biofilm Eradication Concentration, the microbial suspensions were prepared in an analogical manner as described for the assessment of microbiocidal concentration. A total of 100 μL of the suspension was added to the wells of a 96-well microtiter plate and incubated for 24 h at 37 °C. Subsequently, the non-adhered cells were removed by medium aspiration; next the 100 µL of metabolite-containing supernatant was introduced to the wells; thus, the highest concentration of supernatant was 100% (*v*/*v*), while the lowest concentration was 2%. Next, plates were incubated for 24 h/37 °C and subjected to the analogical procedures of TTC introduction or quantitative culturing as procedures performed for planktonic cells. In another version of this investigation line, the medium containing metabolites (not from digested cellulose, but the diffused from the immersed polymer) was applied as antimicrobial; the procedures aiming to assess its potential were performed analogically as in the case of liquid obtained from the dissolved cellulose. As the control of experiment, the MH3 medium collected from suspension of *C. majus* cells (cultured for 28 days) was analyzed using LC-MS/MS technique with regard to content of the main bioactive metabolites. Subsequently, the assessment of MIC and MBEC values toward *S. aureus*, *P. aeruginosa* and *C. albicans* was performed in the analogical manner as it was performed for liquid obtained from digested NBC matrices.”

#### 2.7.3. Viability of Microbial Cells in MH3 Medium

The 10^5^ cfu of *S. aureus, P. aeruginosa*, *C. albicans* and *K. xylinus* were introduced to the MH3 medium and incubated for 1, 3 or 7 days. The microbial suspensions cultivated in the dedicated media (TSB for *S. aureus, P. aeruginosa*, *C. albicans*; H-S for *K. xylinus*) served as control setting. The cells were seeded on the appropriate agar plates and counted at 1, 3 or 7 days of incubation. This technique was repeated three times.

### 2.8. Statistical Evaluation

The results on the anti-inflammatory effect of BNC were expressed as a means ± standard error of mean (SEM). Each sample was tested at least in triplicate in three independent experiments. Firstly, the equality of variances (Levene’s test) for groups and a normality of distribution (Shapiro–Wilk’s test) were verified. Secondly, if neither the homogeneity of variance nor the normality of distribution were detected, a non-parametric test such as Mann–Whitney’s U-test was used in order to establish the statistical significance of differences between means. *p* values below 0.05 were considered statistically significant. All analyses were performed using Statistica 13 software (TIBCO Software Inc., Palo Alto, CA, USA).

## 3. Results

Plant cells viability was estimated after two and four weeks of culture in MH3 liquid medium. The number of live plant cells after two weeks of culture was 1.7 × 10^7^/mL, while after four weeks of culture it was 4.1 × 10^7^/mL (Appendix A). The spatial localization of the plant cells on the carrier was performed by dyeing them with the use of Alamar Blue and revealed their presence mostly in the upper part of carrier (Appendix A). The NBC biosynthesized for 3, 5 and 7 days took the form of cylinders of 50 mm diameter (Figure 1), with an average thickness of 6.71 ± 1.79 mm; 9.28 ± 2.05 mm; 10.42 ± 1.71 mm, respectively, and of average wet mass of 8.73 ± 1.22 g; 14.68 ± 1.45 g; 18.41 ± 1.11 g, respectively. The average mechanical strength of 3-, 5- and 7 day-old BC was 0.51; 0.64 and 0.69 MPa, respectively. The average 3-, 5- and 7 day-old NBC surface area [m^2^/g], pore volume [cm^3^/g], pore diameter [µm] and swelling ratio [%] is presented in Table 1.

The number of *K. xylinus* cells/g in 3, 5 and 7 days-old wet NBC, was 8.8 ± 2.4 × 10^6^; 2.1 ± 9.1 × 10^7^; and 2.2 ± 4.6 × 10^6^, respectively.

Subsequently, we performed Scanning Electron Microscopy to visualize the *C. majus* cells growing on the surface of NBC carriers. The performed analysis revealed the number of aggregated *C. majus* cells (Figure 2A,D) adhered to the NBC surface (Figure 2B,C), confirming usability of NBC in the character of carrier of *C. majus* cells.

In the next investigation line, we performed qualitative phytochemical analyses of liquid released enzymatically from NBC matrices which revealed the high differences in the content of plant metabolites between the type of NBC which served as a carrier for plant cells. A total of 18 compounds were found in the analyzed samples. The richest phytochemical profile was observed for 3- and 5-day-old cellulose containing living *K. xylinus* bacterial cells (NC). Moreover, the accumulation of targeted plant metabolites on these cellulose matrices was maintained throughout the 28 days of culture (Table 2). Regardless of the preparation technique (NC, H_2_O, NaOH), the content of non-alkaloid compounds in 7-day-old celluloses was lower than in 3- and 5-day-old, while isoquinoline alkaloids were present in various, although higher proportions, in most of 7-day-old cellulose samples (Table 2 and Appendix A). On the other hand, in the cellulose which contained bacterial debris (H_2_O) and purified from bacterial cells (NaOH), plant metabolites were detectable only in samples grown for up to two weeks. After this time, the levels of individual compounds were zero or at the limit of detection. The low levels of plant metabolites, in case of these treatments, may have been a result of the weaker growth of *C. majus* cells on 5- and 7-day-old cellulose carriers due to their tighter structure (Table 1). This effect was especially noticeable for cells cultured on NaOH and H_2_O cellulose. In the tested samples, the most abundant group of plant compounds were polyphenols, followed by isoquinoline alkaloids. In the case of non-alkaloid compounds, it was mostly organic acids, such as: malic (retention time – Rt, 1.08 min.), *trans*-aconitic (Rt 1.15), quinic, (Rt 1.30); and benzoic acid derivatives—salicylic (Rt 2.79), protocatechuic (Rt 1.54), tannic (Rt 2.84), and hydroxybenzoic acid (Rt 2.85); as well as hydroxycinnamic acids—*trans*-caffeic (Rt 2.71), *p*-coumaric (Rt 3.97), and chlorogenic (Rt 2.51). One flavonoid (quercetin, Rt 8.34) and one phenolic aldehyde (vanillin, Rt 5.15) (Table 1) were also detected. The following isoquinoline alkaloids were detected: protopine derivatives (protopine, Rt 12.76 and its derivative, Rt 6.76), a protoberberine derivative (coptisine, Rt 7.50) and phenanthridine derivatives (chelidonine, Rt 9.56; chelerythrine, Rt 10.44; sanguinarine, Rt 11.80) (Table 2). 

The most abundant compound was chlorogenic acid. Its content varied between 11.65–12.16 µg/mL for 3-day-old NC cellulose, and 3.77–4.22 µg/mL for 5-day-old NC cellulose. Considerable amount of this compound (7.8 µg/mL) was also detected in 3-day-old H_2_O cellulose. The second most abundant compound was protopine. The highest content of protopine was found, again in 3-day-old cellulose, both NC (5.83–8.5 µg/mL) and H_2_O (7.56 µg/mL) followed by 5-day-old NC cellulose (3.75–5.94 µg/mL) (Table 2 and Appendix A).

The amounts of trans-caffeic acid and quercetin in 3-day-old NC cellulose reached values close to or exceeding 1 µg/mL. Among the remaining compounds, another nine were quantified, although their contents were lower (Table 2). However, the alkaloid profile was richer in the 7-day-old BNCs where non-nitrogenous compounds were either in lower amounts or absent (Appendix A). Based on the phytochemical analysis and the observation of *C. majus* cells growth condition, 3-day-old NC cellulose after 4 weeks of culture were chosen from among others and set for the tests for their bioactivity. Leaving the cultures on 5 and 7-day BNC out from further investigations was based on the symptoms of deterioration observed in these cultures, such as retarded growth and darkening.

Prior performance of the assessment of antimicrobial effectiveness of plant metabolite-enriched BC, we analyzed their effect on the cytokines secretion using LPS-stimulated human neutrophil-based model experiments, because NC version of carrier contained bacterial cells’ leftovers which could induce the activity of the examined immune cells. As shown in Figure 3B,C, TNF-α and IL-8 secretion was decreased in a concentration-dependent manner. In case of IL-1β, the overproduction was observed, but its values decreased along with the higher cellulose concentration used (Figure 3A). The most effective, in terms of TNF- and IL-8 secretion inhibition, was four times diluted cellulose (25%), which released less than 10% of the cytokines, compared to (+) LPS-treated cells. No cytotoxic effect was observed on neutrophils (Figure 3D).

Finally, we analyzed the antimicrobial effect of liquid obtained from 3d NBC (NC, H_2_O, NaOH) carriers against planktonic and biofilm-forming cells of *Staphylococcus aureus*, *Pseudomonas aeruginosa* and *Candida albicans* which represent clinically-relevant Gram-positive- and Gram-negative pathogens and fungus, respectively (Table 3). In turn, in Appendix A we presented the effect of metabolites-enriched medium (in which BNC with *C. majus* was incubated) on above-listed pathogens, while in Appendix A the data showing impact of MH3 medium (applied for plant cells growth) on viability of microbial cells is presented.

The data presented in Table 2 and Appendix A show that BNC, NC-type is not only efficient carrier for *C. majus* cells growth, but also that these cells, immobilized on carrier, are able to produce metabolites in concentrations sufficient to display antimicrobial effect on suspended cells and biofilms formed by wide range (with regard to type of cell wall and origin) of pathogens. The results performed for control setting (medium collected from *C. majus* cell suspension) indicated the presence of bioactive metabolites in it (Appendix A). Nevertheless, their antimicrobial effect was lower (Appendix A) than the effect exerted by digested NBC matrices after co-culture with *C. majus* cells.

## 4. Discussion

The Bio-Nano-Cellulose (BNC) produced by such non-pathogenic bacteria as *Komagataeibacter* species is one of the most promising biopolymers of recent years. The interest in its application in food, pharmaceutical, medicinal and biotechnological industry is constantly growing [31]. It is predicted to increase even more, because one of the main disadvantage of BNC biosynthesis, related with need of expensive culturing medium, was recently overcome by replacing it with various organic, industrial leftovers (such as potato juice) what allowed to produce BNC of properties comparable to these of BNC produced with use of dedicated medium [32]. With regard to material properties, BNC displays a wide spectrum of desirable features, such as high water-related properties and mechanical strength [32]. The production of BNC is, to major extent, environment-friendly; also utilization of BNC does not exert harmful effects on environment, because this organic polymer is fully degraded by soil microorganisms to simple sugars and their derivatives [33]. The advanced technology created features of BNC that enable to use it in medicine and pharmaceutical industry as biotechnology-based products or biopharmaceutics. Our proof of concept based on the biotechnological process of cells immobilization, both plant and bacterial, on the cellulose carrier let to expose the antimicrobial and anti-inflammatory properties of plant metabolites even stronger than when used separately [34]. The carrier (Figure 1) showed differences in physical properties (Table 1) correlated with the biosynthesis time (3, 5 or 7 days). Such investigations, with regard to plant cells incorporation into cellulose network, has not been carried out before, thus at this moment of experiments it was not possible to determine the exact time of biosynthesis that would correlate with the most favorable features of the matrix. Nevertheless, we observed an increasing negative impact of the BNC thickness and tightness resulting from the prolonged days of its synthesis on growth performance of the *C. majus* cells and content of plant metabolites (Table 2), that stays in line with the previously reported data [35]. Using Scanning Electron Microscopy we visualized the *C. majus* cells adhered to BNC surface (Figure 2). We are aware that SEM imaging shows only the surface adhesion of *C. majus* cells to NBC carrier, while providing no data on immobilization pattern occurring potentially within 3D structure of the carrier. Nevertheless, the methodological challenges related with reliable cross-section of NBC carrier of 50 mm diameter and ca. 6 mm thick and its further imaging exceed the scope of this investigation and is one of the limitation of the study, as only the indirect detection of *C. majus* cells in the upper sections of the carrier was performed by means of Alamar Blue dyeing. We plan to resolve this methodological issue by the application of profilometric techniques in the envisaged future study. It can be hypothesized that majority of plant cells (devoid of ability of active movement) will colonize the NBC surface by means of electrostatic/van der Waals interactions, while minority of cells will penetrate, to a specific extent, inside the carrier by means of Brownian movement [36]. Because of the complexity of intertwined relationships between methodological limitations, carriers and cell specific features [37], the utilitarian approach is often proposed—these carriers are perceived as the most appropriate (among those tested), of which application correlates with the production of the highest yield of desired products (such as biotechnologically useful cells or metabolites). Based on this, we performed analysis on the diverse contents of metabolites secreted to BNC by *C. majus* cells (Table 2). These two parameters took the highest values for 3-day-old BNC non-subjected to any process of cleansing. There are at least three variables which could have an impact on such an outcome. First of all, the size of pores and surface area of virtually every biomaterial correlates with cells’ ability to adhere and thrive on it, and, to a certain extent, also with the metabolic condition of the cells (as it was indicated in case of animal-derived cells) [38]. Secondly, the 3-day-old BNC displayed the highest swelling ratio [%]; therefore its ability to trap the metabolites within its structure should be considered the highest among tested BNC matrices. Thirdly, the NC-BNC carrier contained the *K. xylinus* cells which could potentially act as elicitor of plant cell response resulting in the production of specialized metabolites. The bio-elicitation is a well-known phenomenon and in the scientific literature many examples can be found in terms of increasing the production of plant compounds with the use of various microorganism strains, such as β-carboline alkaloids in cell suspension culture of *Peganum harmala* L. [39]. It should be noted that the fewer cells (or their leftovers) in the specific type of carrier, the less production of metabolites (with regard to their concentration) was recorded (Table 1). 

Greater Celandine—*C. majus* is used in phytotherapy for the treatment of gastrointestinal disorders, infections and against warts and other skin protuberances [9]. Our earlier studies have shown that *C. majus* cell cultures are rich in polyphenolic compounds and isoquinoline alkaloids with confirmed antimicrobial and anti-inflammatory properties [8,10,11]. The major metabolites secreted by BNC-immobilized *C. majus* cells were chlorogenic acid, trans-caffeic acid and quercetin; all these compounds are known for their antimicrobial activity and all of them were found in the aerial parts of naturally growing *C. majus* [34]. One of our main concerns, with regard to the applicability of our work, was potential inflammatory effect which could be displayed by NC-type BNC carriers, containing live *K. xylinus* cells. Nevertheless, the results presented in Figure 3 showed no such effect of NC-BNC carriers on human neutrophils. These results stay in line with the recently reported application of *K. xylinus* as probiotic strain and lack of clinical adverse effects from ingestion of these bacteria [40]. However, future investigations on animal models should be performed to obtain the conclusive data on potential adverse effects of BNC matrices containing live *K. xylinus* cells. The highest content of bioactive plant metabolites was recorded in 3-day-old NC- BNC carrier, correlated with the highest antimicrobial effect against Gram-positive, Gram-negative and fungal pathogens (Table 3). The biofilms of tested pathogens displayed higher tolerance to supernatant than their planktonic counter-parts (it was manifested in potential MBEC values above the highest possible-to-achieve concentration of extract), which stays in line with generally acknowledged protective properties of biofilm matrix against antimicrobials [41]. Interestingly, the effect displayed by liquid culture medium MH3 used for cellulose matrices immersion was considerably higher than the effect displayed by supernatant of digested 3-day-old BNC after 4 weeks of incubation with *C. majus* cells (compare Table 3 and Appendix A). Presumably, the porous NBC submerged in MH3 medium, formed micro-environment favorable for retention of the plant cells producing specialized metabolites within the matrix. Conversely, the supernatant obtained after cellulase digestion of *C. majus* metabolite-enriched 3-day-old BNC could have been depleted of plant compounds due to their potential degradation. However, this observation requires further elucidation as it suggests potential interactions between cellulose fibrils and metabolic compounds. Another issue to be addressed in further development of this technology is that the MH3 culture medium was unfavorable with regard to viability of *K. xylinus* in NC-BNC matrices (Appendix A), and also that *C. majus* metabolites were secreted into the direction of the elicitors (microbial cells or cellular left-overs), i.e., into the cellulose carrier. In the third day of incubation, the relative bacterial cell number dropped by 70% and in the seventh day of incubation the number of bacteria was close to null (Appendix A). This trend manifesting in the slow drop of bacteria number suggests that MH3 medium lacked the appropriate nutrient sources for microorganisms. On the other hand, along with the drop of microbes viability, the level of dead cells’ fragments (cell wall, cytoplasm content) was most likely increasing in the environment, additionally eliciting *C. majus* cells to produce antibacterial metabolites. The comparison of antimicrobial effect exerted by the liquid obtained from digested NBC matrices (3-day-old NBC, in particular) vs. medium collected from the suspension of *C. majus* cells showed the higher activity of the first of mentioned liquids. We hypothesize that it may be overlapping result of improved viability and functionality of cells adhered to the carrier [42] vs. suspension cultures as well as of the presence of bacterial elicitors (in NBC matrices) stimulating *C. majus* cells to secrete the metabolites in more active manner. Undoubtedly, future investigations are required to understand this promising (with regard to further medicinal applications) effect.

The data presented in this study showed that BNC can be applied as the carrier for medicinal plant cell culture technology. Such efficient production of the plant biologically active compounds within the bio-nano-cellulose membrane can be used for biopharmaceutical applications. Moreover, we showed that the BNC natural composite did not correlate with an inflammatory effect in vitro, despite the presence of *K. xylinus* bacterial cells, that rather elicited plant cells to produce metabolites of antimicrobial activity. We are aware of laboratory character of our study and necessity of animal studies to get the more detailed insight into the observed phenomena.

Nevertheless, the evidences presented in this work indicate the possibility of application of BNC combined with plant cells producing bioactive substances as bio-composites against microbial pathogens, especially these causing local and surface infections (of skin and wound). For example, using the antimicrobial plant natural products-enriched BNC as an occlusive dressing may be considered.

## 5. Conclusions

We demonstrated that natural matrices such as bacterial nano-cellulose can serve as carriers for plant-derived pharmacologically active substances. The bioactive metabolites exuded from *C. majus* cell cultures were not only retained in the cellulose structure but also did not lose their biological potential. After the *C. majus* cells were introduced onto the BNC, it was verified that the plant cells adhered to the BNC carriers and released pharmacologically active substances. Moreover, various types of NBC cleansing correlated with the metabolites production. However, the depth of plant cells’ infiltration into the BNC matrix would need to be verified by the cross-sections of the BNC carrier through its full thickness. This limitation of the present report will be subject of an upcoming related study.

The mechanisms of stimulation of the plant cells grown on cellulose matrices without removal of the bacteria warrants further research. Understanding the antimicrobial potential of polyphenolic compounds and isoquinoline alkaloids, secreted into bacterial cellulose matrices can serve as a bio-composites for its future use in the treatment of excruciating and often chronic biological skin infections.

## Figures and Tables

**Figure 2 materials-15-00016-f002:**
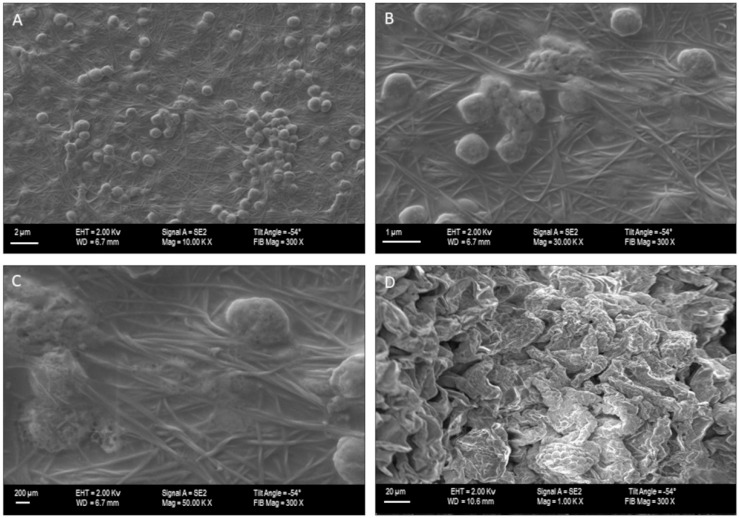
The *C. majus* cellular aggregates (oval shapes) on the 3-day-old NC fibrinous mesh. (**A**): magnification 10,000×; (**B**): 30,000×; (**C**): 50,000×, (**D**): close-up on *C. majus* cells, 100,000×. Microscope SEM Zeiss Auriga 60.

**Figure 3 materials-15-00016-f003:**
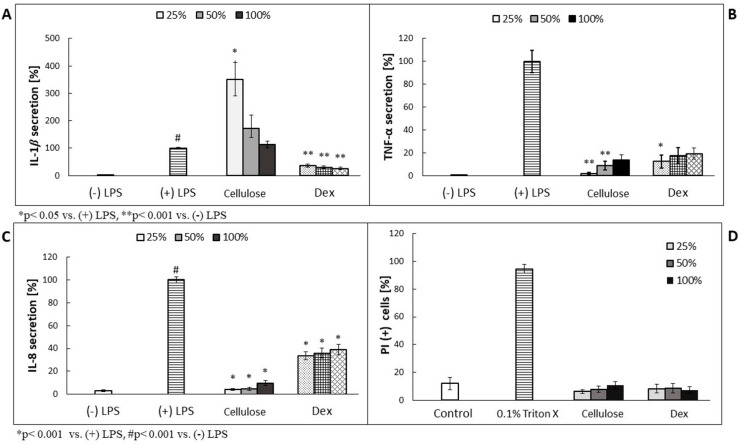
(**A**–**C**): The effect of 3-day-old NC cellulose containing *C. majus* cells on neutrophil IL-1β, TNF-α and IL-8 release by LPS—stimulated (100 ng/mL) cells ((+) LPS). Dex—dexamethasone at concentrations of 0.25, 0.5 and 1.0 µM; (**D**): The cytotoxic effect of 3-day-old NC cellulose containing *C. majus* cells (after four weeks of culture) on neutrophils expressed as a percentage of PI (+) cells; Control, (+) LPS; cellulose = 3-day-old NC cellulose containing *C. majus* cells (after four weeks of culture) The data are expressed as the mean ± SE from three donors assayed in triplicate. Statistical significance of the cellulose versus the stimulated control at *p* ˂ 0.001 vs. (-) LPS is marked by ** (**A**,**B**), vs. (+) LPS by * (**C**), and vs. (-) LPS by # (**C**); at *p* ˂ 0.005 vs. (+) LPS is marked by * (**A**,**B**) (Mann–Whitney’s U- U-test).

**Table 1 materials-15-00016-t001:** NBC carriers (3, 5 and 7 days old) characteristics.

Carrier/Feature	Surface Area [m^2^/g]	Pore Volume [cm^3^/g]	Pore Diameter [µm]	Swelling Ratio [%]
3 days old NBC	10.52 ± 0.77	3.88 ± 0.12	0.086 ± 0.1	397 ± 55
5 days old NBC	8.92 ± 3.89	3.63 ± 1.97	0.022 ± 0.002	273 ± 18
7 days old NBC	8.4 ± 0.62	3.26 ± 0.08	0.018 ± 0.001	232 ± 52

**Table 2 materials-15-00016-t002:** LC-MS/MS analysis of the content of the detected compounds and their retention times (Rt) in 3-day-old cellulose (NaOH, H_2_O, NC) with *C. majus* cells cultured on MH3 medium for 14, 21 and 28 days.

Compound	Rt [min]	NC [µg/mL ± SE]			H_2_O [µg/mL ± SE]	NaOH [µg/mL ± SE]
		14 Days	21 Days	28 Days	14 Days *	14 Days *
malic acid	1.08	p	p	p	p	p
*trans*-aconitic acid	1.15	p	nd	nd	p	p
quinic acid	1.30	nd	p	p	p	p
protocatechuic acid	1.54	nd	p	p	nd	nd
chlorogenic acid	2.51	nd	11.65 ± 0.52	12.166 ± 1.314	7.8 ± 0.40	LOQ
trans-caffeic acid	2.71	nd	1.4 ± 0.144	1.550 ± 0.100	0.411 ± 0.031	LOQ
salicylic acid	2.79	nd	0.058 ± 0.008	0.075 ± 0.000	LOQ	nd
tannic acid	2.84	nd	1.0 ± 0.100	nd	LOQ	nd
hydroxybenzoic acid	2.85	nd	0.05 ± 0.00	0.075 ± 0.00	LOQ	nd
*p*-coumaric acid	3.97	nd	nd	0.150 ± 0.027	LOQ	nd
vanillin	5.15	nd	nd	0.350 ± 0.028	LOQ	nd
protopine derivative	6.75	p	p	p	p	p
coptisine	7.50	LOD	nd	nd	LOQ	LOQ
quercetin	8.34	LOD	1.05 ± 0.02	0.966 ± 0.016	LOQ	nd
chelidonine	9.56	LOD	LOQ	LOQ	LOD	nd
chelerythine	10.44	nd	nd	nd	0.125 ± 0.029	nd
sanguinarine	11.80	LOQ	0.05 ± 0.00	LOD	0.022 ± 0.002	nd
protopine	12.76	6.25 ± 0.75	8.50 ± 0.764	5.833 ± 0.601	7.566 ± 0.929	LOQ

p—present, where identification was based on mass spectra with no reference substances; nd—not detected; * after 14 days, the cell cultures showed symptoms of deteriorating, therefore, these cultures have not been further analyzed; LOQ—limit of quantitation, calculated as a signal to noise ratio of 10; LOD—limit of detection, calculated as a signal to noise ratio of 3 (see the Materials and Methods).

**Table 3 materials-15-00016-t003:** Minimal Antimicrobial Concentration and Minimal Biofilm Eradication Concentration of liquid obtained from NBC matrices.

	Minimal Microbiocidal Concentration *v*/*v* [%]			Minimal Biofilm Eradication Concentration *v*/*v* [%]		
	*S. aureus*	*P. aeruginosa*	*C. albicans*	*S. aureus*	*P. aeruginosa*	*C. albicans*
3d BC-NC	12.5%	25%	25%	50%	100%	50%
3d BC -H_2_O	50%	25%	50%	100%	above tested range of con-centrations	above tested range of con-centrations
3d BC-NaOH	50%	50%	50%	above tested range of concentrations	above tested range of concentrations	above tested range of concentrations

## Data Availability

The data presented in this study are included in the article and Appendix A. Further data generated in this research are available on request from the corresponding author. These data are not publicly available due to the funding bodies rules.

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
