# Peer review of "Bacterial Nanocellulose Fortified with Antimicrobial and Anti-Inflammatory Natural Products from Chelidonium majus Plant Cell Cultures"

_materials, 2021, doi:10.3390/ma15010016_

Round 1

Reviewer 1 Report

The reviewed paper focused on the use of bacterial nanocellulose containing plant producing antimicrobial compounds. BNC that was synthesized for 3, 5 and 7 days by K. xylinus were tested. The BNC either still contained K. xylinus cells, the cells were killed by autoclaving or removed by treatment with NaOH. C. majus plant cells were incubated in the presence of these three different types of BNC and the antimicrobial properties tested using model pathogenic bacteria and fungus. Anti-inflammetory response of the above samples were also tested using model immuno assays. 

Minor Comments:

Please reorganize the order in which the tables are presented. Some are presented out of sequence. 

Label the magnification in your Figure 1S so it is visible. It is too small to see.

Define LOQ abbreviation in your Table legends.

In the discussion, it would help the reader if it is explained why in the antimicrobial assays some of the results were outside the range of the analysis (Tables 3 and 1S).

Also, why is there a difference in the antimicrobial effect between single cultures of the pathogenic bacteria tested vs the biofilm form? (Tables 3 and 1S). This seems like an important point to discuss further.

Table 1S is not discussed at all and should be referred to in the discussion.

Why was a statistical cut off of 0.001 used in one case vs 0.005 when assessing the results presented in Figure 3? Also, briefly explain in your method section why you selected the Mann Whitney U-test so that future readers can understand the logic behind this and consider using your study as a reference for this type of statistical analysis. 

Author Response

The response to this Reviewer's comments are attached as a PDF file

Reviewer 2 Report

Dear authors,

The revised manuscript entitled " Bacterial nanocellulose fortified with antimicrobial and anti-inflammatory natural products from Chelidonium majus plant cell 3 cultures " evaluated the potential of Bacterial-Nano-Cellulose as a carrier system for cell cultures of Chelidonium majus as a medicinal plant producing antimicrobial compounds. The authors did a lot of work, and the results in the current study are of interest and a somewhat interesting research project. However, there are major issues that should be considered before this manuscript can be accepted.

-The present introduction insufficiently explains the scope of the study. Very important literature related hypotheses, potential mechanisms about BNC as a carrier of cells derived from plants and indicators monitored during this study are missing. Instead, a lot of vague sentences was repeatedly used in some parts of the introduction.

Methodology

The 2.3.2. subsection related to the weight of wet BC carriers is better to combine into the 2.3.1. subsection

- the authors mentioned that "The mechanical strength was determined using the MTS Synergie 100® machine 140 (MTS System Corp, MA, USA)." However, they didn’t show any related results or data.

-The section on " plant cell cultures on the BNC matrices" is insufficiently described, and more details should be introduced. The processes schematic is highly recommended to add by authors.

-In SEM images, it is expected and highly recommended to show the cross-sections beside the surface-sections of the 3, 5, and 7 -day-old plant cell cultures on BNC matrices.

-The quality of Figure 3 should be increased.

-In such a research paper, the antimicrobial effect of the free C.majus metabolites-enriched medium is expected to be tested and compared to the liquid obtained from NBC matrices.  

Conclusions section

-The novelty of this work should be focused and the authors should emphasise the major conclusion of the main purpose of the current study.

Author Response

(The authors gave the same response as above.)

Reviewer 3 Report

In this work the authors  investigate the feasibility of using bacterial nanocellulose as a carrier of plant cells. The manuscript is well organized and presented with many experimental results. The conclusions are well supported by the experimental results

Author Response

(The authors gave the same response as above.)

Round 2

Reviewer 2 Report

A considerable effort was made by authors to revise the manuscript and addressed all issues sufficiently. However, one minor comment is that the authors are recommended to point out the issue related to the cross-sections of BNC matrices embedded with plant cell cultures observation and the depth of plant cell adhesion as limitation point of this study which can be further observed in upcoming related studies in the conclusion section. Also, it is recommended to refer to the Figure 1. showed in the response report "Vertical cross-section BNC containing C.majus cells observed after dye application" in the main text as a preliminary test. After address these minor comments, I recommend this manuscript for publishing in Materials.

Author Response

The reply to the reviewer's comments is in the attached PDF file.
